# Predicting Leaf Nitrogen Content in Cotton with UAV RGB Images

**Jinmei Kou** [1,2,†], **Long Duan** [3,†], **Caixia Yin** [1], **Lulu Ma** [1], **Xiangyu Chen** [1], **Pan Gao** [3,*] **and Xin Lv** [1,*]

1   The Key Laboratory of Oasis Eco-Agriculture, Xinjiang Production and Construction Group, College of Agriculture, Shihezi University, Shihezi 832003, China; joujinmei@shzu.edu.cn (J.K.); yincaixia@stu.shzu.edu.cn (C.Y.); malulu@stu.shzu.edu.cn (L.M.); 20192112001@stu.shzu.edu.cn (X.C.)
2   College of Mechanical and Electrical Engineering, Shihezi University, Shihezi 832003, China
3   College of Information Science and Technology, Shihezi University, Shihezi 832003, China; duanlong@stu.shzu.edu.cn
*   Correspondence: gp_inf@shzu.edu.cn (P.G.); luxin@shzu.edu.cn (X.L.)
†   These authors contributed equally to this work.

**Abstract:** Rapid and accurate prediction of crop nitrogen content is of great significance for guiding precise fertilization. In this study, an unmanned aerial vehicle (UAV) digital camera was used to collect cotton canopy RGB images at 20 m height, and two cotton varieties and six nitrogen gradients were used to predict nitrogen content in the cotton canopy. After image-preprocessing, 46 hand features were extracted, and deep features were extracted by convolutional neural network (CNN). Partial least squares and Pearson were used for feature dimensionality reduction, respectively. Linear regression, support vector machine, and one-dimensional CNN regression models were constructed with manual features as input, and the deep features were used as inputs to construct a two-dimensional CNN regression model to achieve accurate prediction of cotton canopy nitrogen. It was verified that the manual feature and deep feature models constructed from UAV RGB images had good prediction effects. $R^2 = 0.80$ and RMSE = 1.67 g kg$^{-1}$ of the Xinluzao 45 optimal model, and $R^2 = 0.42$ and RMSE = 3.13 g kg$^{-1}$ of the Xinluzao 53 optimal model. The results show that the UAV RGB image and machine learning technology can be used to predict the nitrogen content of large-scale cotton, but due to insufficient data samples, the accuracy and stability of the prediction model still need to be improved.

**Keywords:** UAV-RGB image; image analysis; leaf nitrogen content; cotton; machine learning

## 1. Introduction

The precise application of nitrogen fertilizer based on crop growth is a current hotspot in research on agriculture [1]. Traditional nitrogen fertilizer management is based on field soil sampling and testing or plant nutrition analysis, with an advantage of high prediction accuracy for crops at a certain sampling point. However, due to the high cost, it cannot meet the need of accurate monitoring of crop nutrients in large areas and multiple periods. Compared with traditional methods, image-based plant nutrition detection has the advantages of fast sampling speed, strong real-time detection, and high prediction accuracy [2].

By combining remote sensing technologies, image-based monitoring can realize large-scale crop growth monitoring in the field. Lee et al. [3] found that the $R^2$ reached 0.85, and the root mean square error reached 4.52 g m$^{-2}$ through combining unmanned aerial vehicle (UAV) multi-spectral images with machine learning methods in predicting the corn canopy nitrogen content in the field. However, remote sensing image data contains a large amount of redundant information [4]. Extracting useful data from the complex image data to construct a prediction model is always faced with a tough technical challenge. The present methods for feature extraction from image data mostly focus on color and texture [5].

Feature extraction is the first step in constructing a nitrogen prediction model, and further screening and fusion of the extracted features need to be combined with machine learning methods, including data analysis methods such as classification, clustering, and regression. Machine learning methods have obvious advantages in discriminating model construction methods [6].

As the scale increases, the height of data collection also increases, but the prediction accuracy of the model based on the data collected by the existing image acquisition equipment is not improved with the increase in height, which limits the application of remote sensing technology in the monitoring of crops. Deep learning, a sub-branch of machine learning, is a learning process that uses deep neural networks in extracting features. Deep learning shows high prediction accuracy in practice and can obtain very complex underlying patterns for the data. It is especially suitable for large data sets and high-dimensional data sets. The hidden layer of the network significantly reduces the demand for features engineering. These characteristics entail unique deep learning advantages in crop monitoring [7]. With the help of deep learning, the features that can be used for crop monitoring can be effectively extracted [8].

Taking cotton plants in the field as the research subject, this study constructed a nitrogen prediction model based on the cotton canopy data collected by using the UAV and the appropriate feature screening and extraction methods. The aim of this study was to provide a technical measure for the realization of fast, economical, non-destructive, and high-efficient cotton nitrogen estimation in cotton fields and build a theoretical basis for remote sensing technology for crop monitoring.

## 2. Materials and Methods

### 2.1. Experimental Design

This experiment was carried out in the suburbs of Shihezi City, Xinjiang, China (44°18′52.81″ N, 85°58′48.27″ E) in 2019. Cotton was sown in April 2019 and harvested in October 2019. The water and fertilizer were applied through drip irrigation under the mulch. The film width was 2.05 m. Six rows were under each film. The wide row spacing between two rows was 0.66 m, and the narrow row spacing was 0.10 m. Wide and narrow rows were arranged alternately. The seeding density was about 110,000 plants/hm$^2$ (Figure 1).

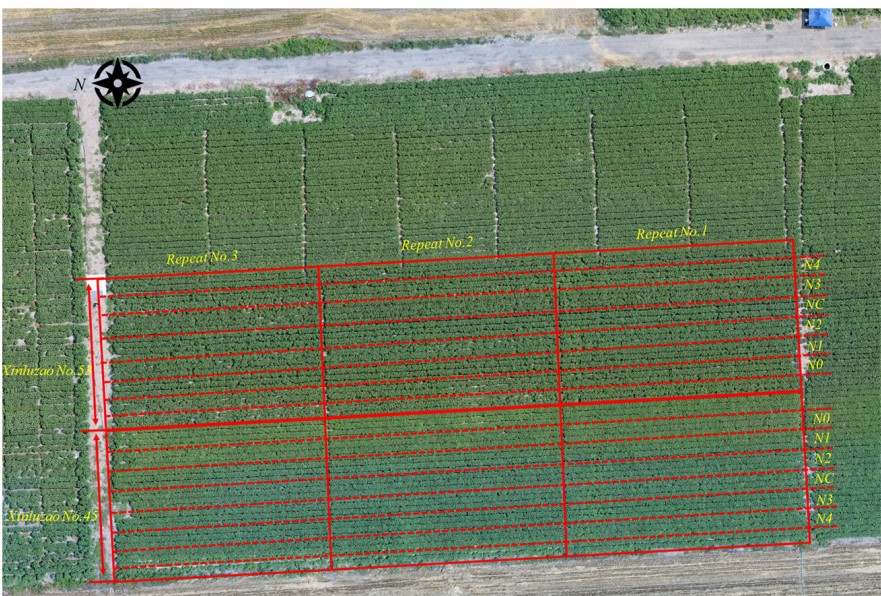

**Figure 1.** Overview of experimental site.

The soil type of the test site was irrigated gray desert soil, with organic matter, available nitrogen, available phosphorus, and available potassium content of 11.75 g kg$^{-1}$, 0.45 mg kg$^{-1}$, 0.39 mg kg$^{-1}$, and 375.21 mg kg$^{-1}$. To improve the generalization and accuracy of the prediction model, two varieties of cotton Xinluzao 45 and 53 were selected as the research subjects in this study, and six nitrogen fertilizer gradients (0 kg hm$^{-2}$ (N0), 120 kg hm$^{-2}$ (N1), 240 kg hm$^{-2}$ (N2), 330 kg hm$^{-2}$ (NC), 360 kg hm$^{-2}$ (N3), and 480 kg hm$^{-2}$ (N4)) were set. Each gradient had three repetitions, and a total of 36 plots was set. The area of each plot was 62.5 m$^2$ (2.5 × 25) (Figure 1). The applied nitrogen fertilizer was urea, which was dissolved in a differential pressure tank and applied through drip irrigation.

### 2.2. Data Acquisition

RGB image data of the cotton canopy was collected five times from June to August in 2019 (June 28, July 6, July 16, July 28, and August 7) using digital cameras on a four-rotor UAV (Phantom 4 advance, DJI, China). The takeoff weight of the UAV was 1380 g, the maximum flight speed was 20 m/s, and the flight took about 30 min. The camera was a 1-inch CMOS camera with an effective pixel of 20 million. In the process of data acquisition, the flight altitude was 20 m, the ground resolution was about 0.5 cm, the overlap rate of heading and side direction was set to 75%, and the collected data were saved in the *.TIF format. Pix4D ag (Pix4D, Lausanne, Switzerland) software was used for UAV image stitching (Figure 1). To minimize the influence from the difference in light intensity at different sampling times on image quality and improve the comparability of the images, the time of 12:00–14:00 with strong light and relatively stable solar altitude angle was selected for image acquisition.

Three cotton samples were selected on the side and middle row of the membrane in each plot for the nitrogen content measurement. The root, stem, and leaves were separated and dried in an oven until constant weight. The dry weight of each organ was weighed. The dried stem and leaves were smashed separately and digested with concentrated sulfuric acid (98%) to determine the leaf nitrogen content in cotton by the Kjeldahl method (Figure 2). The average value of two sampling points in each plot was used for analysis.

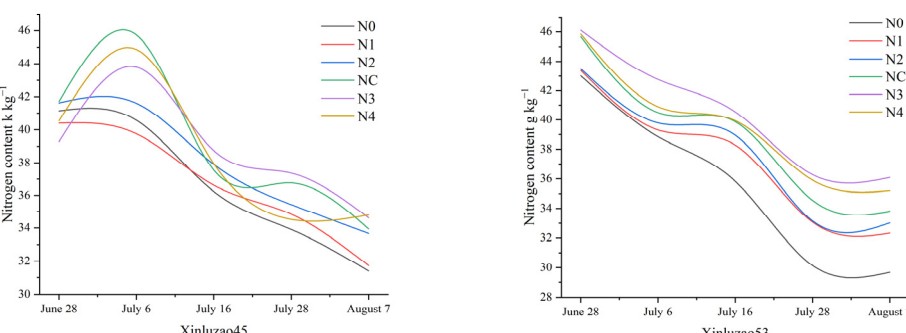

**Figure 2.** Nitrogen status of Xinluzao 45 and Xinluzao 53.

The images collected from one plot in each sampling were taken as one sample. A total of 180 samples was collected. The data set was divided into the training set and the test set at a ratio of 2:1. For each nitrogen application gradient, two replicates were set as the training set, and one was set as the test set (Table 1).

**Table 1.** Data set division.

| Variety | Data Set | June 28 | July 6 | July 16 | July 28 | August 7 | Total |
|---|---|---|---|---|---|---|---|
| Xinluzao45 | Training set | 12 | 12 | 12 | 12 | 12 | 60 |
| | Test set | 6 | 6 | 6 | 6 | 6 | 30 |
| Xinluzao53 | Training set | 12 | 12 | 12 | 12 | 12 | 60 |
| | Test set | 6 | 6 | 6 | 6 | 6 | 30 |

### 2.3. Image Preprocessing

ENVI 5.3 software (ENVI, Exelis Visual Information Solutions, Boulder, CO, USA) was used to divide the image data collected in each sampling. The ROIs of the target plots were manually marked, and the mask was attached to output the image data of the target plots. The marking of the soil pixels in the intervals between rows as plant regions caused by the slight error in the selection of ROIs, and the appearance of a large amount of soil background in the early growth stage of cotton will affect the detection accuracy of the model. Therefore, this study adopted a threshold segmentation method to remove noise such as soil in the target area and only retained the cotton canopy pixels.

### 2.4. Feature Extraction

Traditional manual scene classification includes low-level features and intermediate features. Low-level features mainly refer to features that can be obtained through simple operations based on the image itself, such as color, texture, shape, etc. These features are often those with distinguishing features extracted by imitating human vision. Intermediate features refer to those obtained by multi-feature fusion based on low-level features, which are mainly about the diverse fusion of color and texture.

In this study, 46 features including 25 visible-band vegetation indexes (Table 2), the HSI color model, and image texture were used as manual features. Texture features included mean, standard deviation (std), angular second moment (Asm), entropy (Ent), contrast (Con), and inverse differential moment (Idm) under gray level co-occurrence matrices of $0°$, $45°$, $90°$, and $135°$. Hue (H), saturation (S), and intensity (I) were used in the HSI color model to represent images, which was more in line with the way people describe and interpret colors.

**Table 2.** Visible-band vegetation indexes.

| Vegetation Index | Variable | Vegetation Index | Variable |
|---|---|---|---|
| $R, G, B$ | V1, V2, V3 | $(g^2 - r^2)/(g^2 + r^2)$ | V15 [9] |
| $r, g, b = R/N, G/N, B/N$ $N = R + B + G$ | V4, V5, V6 | $(g^2 - b \times r)/(g^2 + b \times r)$ $(g - r)/(g + r)$ | V16 [9] V17 |
| $r/g$ | V7 | $(2g - r + b)/(2g + r + b)$ | V18 |
| $g/b$ | V8 | $(2g - r - b)/(2g + r - b)$ | V19 [10] |
| $r/b$ | V9 | $1.4r - g$ | V20 [11] |
| $r - b$ | V10 | $2g - r - b$ | V21 |
| $r + b$ | V11 | $V21 - 1.4r - g$ | V22 [12] |
| $g - b$ | V12 | $0.441r - 0.881g + 0.3856b + 18.78745$ | V23 [13] |
| $(r - b)/(r + b)$ | V13 | $(g - r)/(g + r - b)$ | V24 [14] |
| $(r - g - b)/(r + g)$ | V14 | $(g - b)/(r - g)$ | V25 |

Deep features are usually deep and abstract features of images selected by the neural network model. It has the characteristics of no manual participation and less influence from illumination and posture, but the expression of features cannot be clearly identified. In this study, convolutional neural networks (CNNs) were used for deep feature extraction.

### 2.5. Feature Dimensionality Reduction

In this study, 46 manual features were extracted, but the redundant information and noise could cause errors in model construction and reduce the accuracy. Therefore, data dimensionality reduction was used to extract the internal essentials of the features, reduce the errors caused by redundant information and noise, and facilitate calculation and visualization. Data dimensionality reduction included feature extraction and feature selection. Feature extraction transformed high-dimensional feature vectors into low-dimensional feature vectors through mapping (transformation). Feature selection realized data dimensionality reduction by selecting some of the most representative and good-performance features from the original features.

Partial least squares-based dimension reduction (PLSDR) is a data dimensionality reduction method based on feature extraction [15]. Through modeling with partial least squares to extract features, the potential components that could represent the original data space were obtained, and the data dimensionality reduction was achieved. Pearson, a commonly used data dimensionality reduction method based on feature selection, was used to select features by calculating the correlation and coefficient. PLSDR and Pearson were used for manual features dimensionality reduction in this study.

*2.6. Model Construction*

Multiple linear regression (LR), a model analyzing the linear relationship between feature variables and predicted values, was used to construct the model. Support vector machine (SVM) is a generalized linear classifier for nonlinear classification by the kernel method [6,16], and its decision boundary is the maximum margin hyperplane for solving learning samples. CNN is constructed by imitating the visual perception mechanism. The parameter sharing of the convolution kernel in the hidden layer and the sparsity of inter layer connection enable the CNN to learn the features of pixels with less computation. CNN can not only extract deep features by increasing the depth of the network [16,17] but also avoid complex feature engineering. Therefore, CNN is widely used in coping with problems in regression and classification [18,19]. In this study, LR and SVM were used for manual feature model construction, and two CNN structures (Figure 3) were used for manual feature and deep feature model construction, respectively. The one-dimensional (1D) CNN (Figure 3a) included an input layer, three 1D convolution layers (1D-CNN), a full connection layer (FC), and an output layer. The 1D-CNN included a 1D convolution structure (Conv1D), a rectified linear unit activation function (ReLU), batch normalization (BatchNorm), and max pooling (MaxPool). The number of convolution kernels of the three 1D-CNNs was 32, 64, and 128. The two-dimensional (2D) CNN (Figure 3b) included an input layer, four 2D convolution layers (2D-CNNs), an FC, and an output layer. The 2D-CNN included a 2D convolution structure (Conv2D), ReLU, BatchNorm, and MaxPool. The number of convolution kernels of the four 2D-CNNs was 32, 64, 128, and 256.

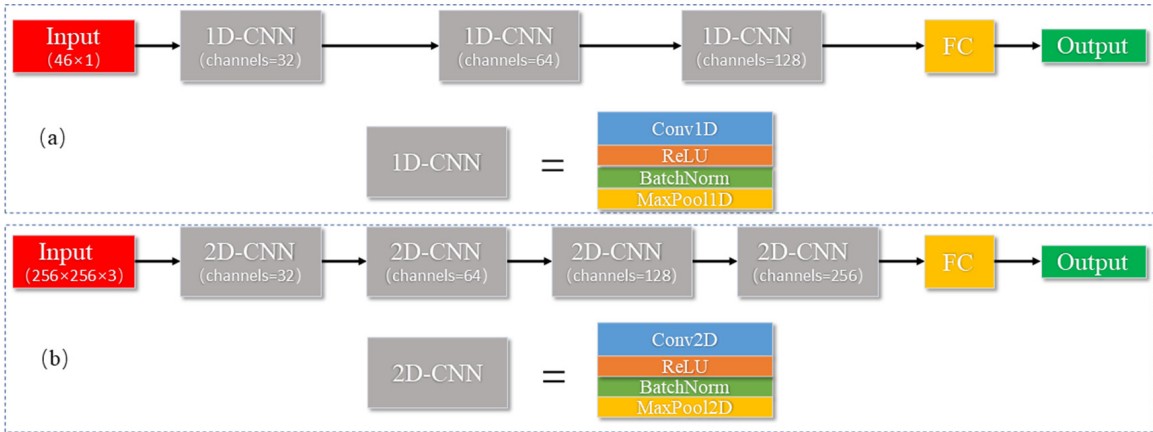

**Figure 3.** CNN structures: (**a**) 1dcnn structure, (**b**) 2dcnn structure.

## 3. Results and Discussion

### 3.1. Analysis of Data Dimensionality Reduction Results

Partial least squares analysis was carried out for the data of the two cotton varieties in the experiment. For Xinluzao 45, when ten principal components were remaining, the contribution rate of principal components was 99.90%, and the root mean square error was the lowest (Figure 4a). For Xinluzao 53, when eight principal components were remaining, the root mean square error was the lowest, and the contribution rate of principal components was 99.93% (Figure 4b).

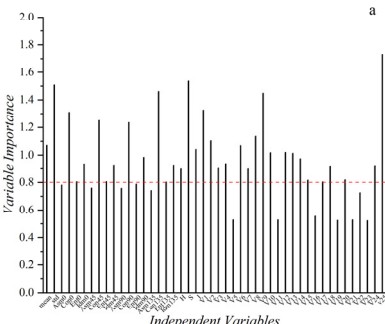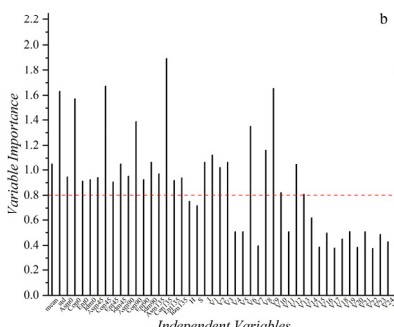

**Figure 4.** Importance of partial least squares variables: (**a**) Xinluzao 45, and (**b**) Xinluzao53.

Variables such as standard deviation and contrast in texture features, saturation in HIS color space, and V9 and V25 in the color index were of great importance. When analyzing the data of the two varieties, the same variables had different contribution rates to the model, but the set of important variables had high similarity (Figure 4). This is because there are phenotypic similarities among different cotton varieties, which can be used as a reference for model generalization.

Pearson correlation analysis was carried out on the 46 features extracted from the image data of the two varieties of cotton, at the significance level of 0.01. There were 25 significantly correlated features extracted from Xinluzao 45 (Figure 5a) and 28 significantly correlated features extracted from Xinluzao 53 (Figure 5b).

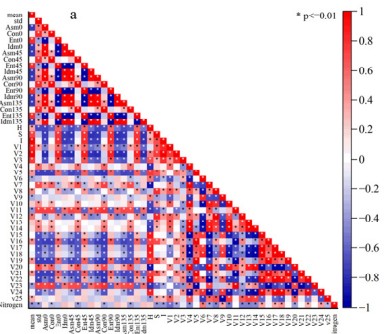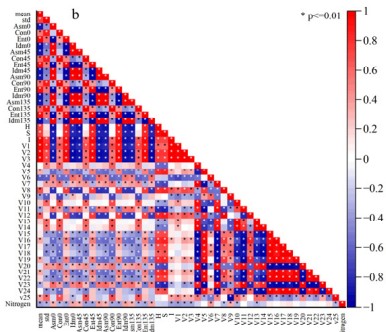

**Figure 5.** Pearson correlation analysis: (**a**) Xinluzao 45, (**b**) Xinluzao 53.

There were strong correlations among features, especially the features for Xinluzao 53, such as texture features and color indexes of V15–V22, indicating that the feature data had redundancy. If a large number of highly correlated features was selected, it would reduce the effect of model fitting (Figure 5). Therefore, the features obtained through correlation analysis were sorted during model training, and the optimal feature set was selected after model verification.

### 3.2. Manual Feature Regression Model

3.2.1. Results of Traditional Machine Learning Methods

The partial least squares regression model was constructed based on the features obtained by the partial least squares method. The LR and SVR models were constructed based on the feature sets screened by Pearson correlation analysis. The Python 3.6.4 programming language was used in modeling. When constructing the PLSR model, the optimal data sets obtained by partial least squares dimensionality reduction were input, in which Xinluzao 45 contained 10 principal components, and Xinluzao 53, which contained 8 principal components. When modeling with the features screened by Pearson correlation analysis, the features were sorted according to the correlation strength, and LR and SVR models were constructed with the data sets of different sizes (2, 3, . . . , 23) to screen the optimal model. After verification, for Xinluzao 45, 14 features were used to construct the LR

model, and 16 features were used to construct the SVR model. For Xinluzao 53, 9 features were used to construct the LR model, and 13 features were used to construct the SVR model (Figure 6).

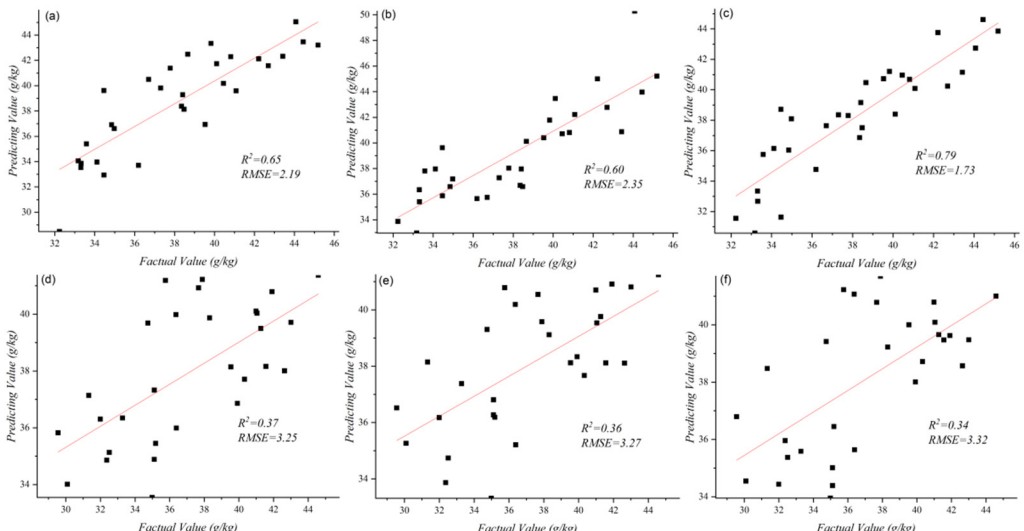

**Figure 6.** Prediction results: (**a**–**c**) Prediction results of Xinluzao 45 (LR, SVR, and PLSR); (**d**–**f**) prediction results of Xinluzao 53 (LR, SVR, and PLSR). The horizontal axes and vertical axes are the factual and model-predicted nitrogen content values of cotton, respectively.

The model fitting for Xinluzao 45 was better than that of Xinluzao 53 (Figure 6). This was because when collecting UAV remote sensing data, the growth of Xinluzao 45 was better than that of Xinluzao 53, and the complexity of each plot in the image data of the latter was higher than that of the former. For example, compared with Xinluzao 45, Xinluzao 53 had less dense vegetation and more exposed bare soil (Figure 1), which brought difficulties to evaluate the leaf nitrogen level of the whole site. However, after the removal of noise and feature screening, the correlation coefficient was still maintained at 0.37.

### 3.2.2. Results of 1D CNN

To test the superiority of the multi-layer convolution network for feature extraction, this study took 46 features of the two varieties as inputs and constructed a 1D CNN model. The framework used was pytorch1.5.0, the loss function was MSELoss, the learning rate was 0.001, and the number of iterations was 200.

The predicted results of 1D CNN were better than those of traditional machine learning methods (Figure 7), and there was no need for feature dimensionality reduction. To verify the effectiveness of CNN feature extraction, the saliency map was used to visualize the significance of CNN features. In Figure 8, it can be seen that the features sensitive to the model were similar to the important features of feature dimensionality reduction screening, and CNN could extract deeper fused features, which led to the good performance of CNN.

### 3.2.3. Results of Deep Feature Regression Model

The 1D CNN model constructed using manual features had reliable performance in the prediction of leaf nitrogen content of the two varieties. Therefore, this study used the 2D CNN model for direct extraction of the deep features of UAV remote sensing images, which reduced the steps of feature engineering. The model was constructed with the pytorch1.5.0 framework, the loss function was MSELoss, the learning rate was 0.0005, and the number of iterations was 200. GeForce GTX 1080 Ti graphics cards were used for acceleration. The model prediction results are shown in Figure 9. The $R^2$ of the model for Xinluzao 45 was 0.79, and that for Xinluzao 53 was 0.418. The 2D CNN model performed well in both the training set and the test set, especially in the training set, and the $R^2$ values of both

varieties were greater than 0.9, which proved that the 2D CNN was effective for the deep features extracted. We will increase the number of training data in subsequent experiments to improve the accuracy and stability of the results.

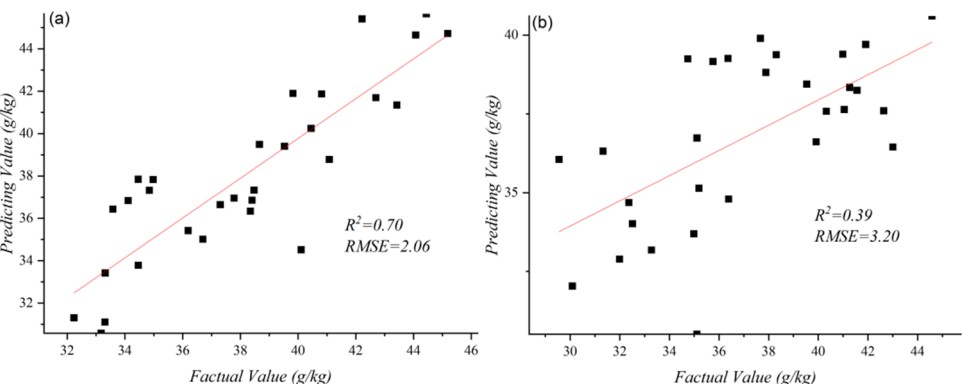

**Figure 7.** The 1D CNN prediction results: (**a**) Xinluzao 45, (**b**) Xinluzao 53. The horizontal axes and vertical axes are the factual and model-predicted nitrogen content values of cotton, respectively.

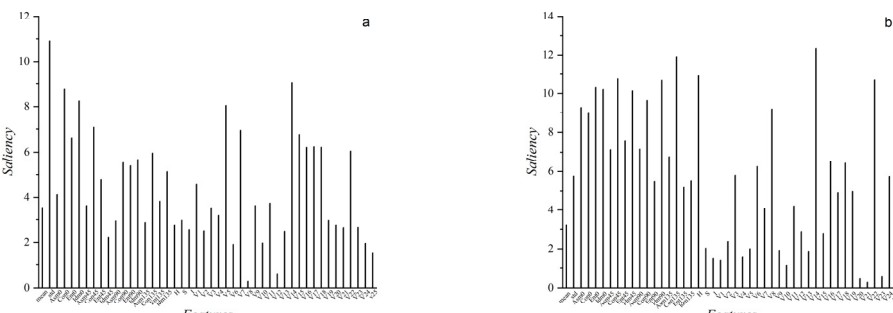

**Figure 8.** The 1D CNN saliency map: (**a**) Xinluzao 45, (**b**) Xinluzao 53. The horizontal axes are the 46 manual feature variables extracted in this study, and the vertical axes are the feature saliency obtained by the saliency map method.

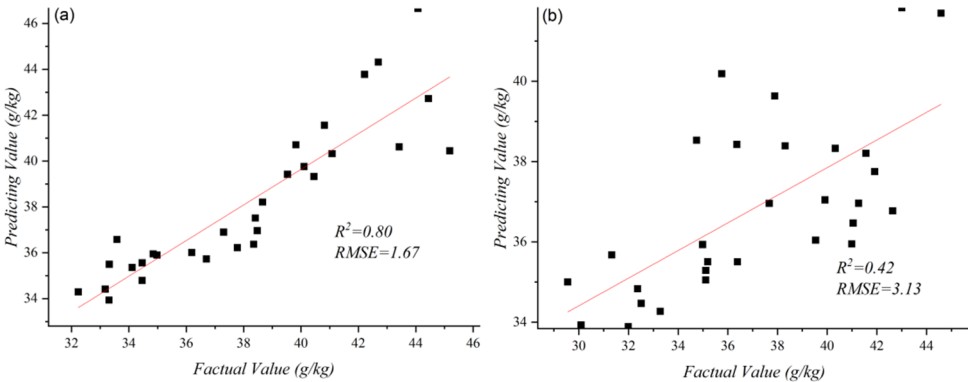

**Figure 9.** The 2D CNN prediction results: (**a**) Xinluzao 45, (**b**) Xinluzao 53. The horizontal axes and vertical axes are the factual and model-predicted nitrogen content values of cotton, respectively.

## 4. Conclusions

In this study, UAV-based RGB remote sensing image data were collected for the prediction of cotton leaf nitrogen content. Through the combination of feature extraction, feature dimensionality reduction, and machine learning model construction, the prediction of cotton nitrogen content based on manual features was realized. Reliable prediction accuracies were obtained for the models for both cotton varieties (Xinluzao 45, $R^2 = 0.79$, RMSE = 1.73 g kg$^{-1}$; Xinluzao 53, $R^2 = 0.39$, RMSE = 3.20 g kg$^{-1}$). By constructing a

CNN model, the prediction of cotton nitrogen content based on deep features was realized (Xinluzao 45, $R^2$ = 0.80, RMSE = 1.67 g kg$^{-1}$; Xinluzao 53, $R^2$ = 0.42, RMSE = 3.13 g kg$^{-1}$). The CNN model does not need the construction of feature engineering, and the extracted deep features are highly effective in model prediction. Therefore, it is feasible to predict cotton nitrogen content using UAV-based RGB remote sensing images, but it is necessary to further improve the generalization of prediction models for different varieties. In future research, we will pay attention to more cotton varieties, planting areas, and growth periods; continue to expand data samples and improve the cotton nitrogen content prediction model; and try to apply this method for the nutrition detection of other crops.

**Author Contributions:** Conceptualization, J.K. and X.L.; methodology, L.D.; software, L.D.; validation, C.Y., L.M. and X.C.; formal analysis, J.K.; investigation, J.K.; resources, C.Y.; data curation, J.K. and L.D.; writing—original draft preparation, J.K. and L.D.; writing—review and editing, L.D.; visualization, J.K.; supervision, X.L. and P.G.; project administration, X.L. and P.G.; funding acquisition, X.L. All authors have read and agreed to the published version of the manuscript.

**Funding:** This work was supported by the auspices of the National Natural Science Foundation of China (42061058), and the Major Science and Technology Project of XINJIANG Production and Construction Corps (2020AB005).

**Data Availability Statement:** Data are not available online; those interested in the data can contact the authors via the email addresses provided.

**Conflicts of Interest:** The authors declare no conflict of interest.

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
