# Peer review of "Predicting Leaf Nitrogen Content in Cotton with UAV RGB Images"

_sustainability, doi:10.3390/su14159259_

Round 1

Reviewer 1 Report

Line 14: Typo “Abstract: Abstract:”

Line 19: Typo “construct-ed”

Introduction

Line 31: Typo “ad-vantage”

Figure 1: Left maps are not necessary, aerial images with experiment setup and coordinates are enough. It would be better to enhance the quality of figures and labels of the experiment setup.

Line 111: Could you explain more how to get those 180 samples. I am confused with table 1. Last column does not sum 180. It is 60+30+90+30.

2.4 Feature extraction: Table 2 explains the 25 variables, and it sets on the variables columns the nomenclature used in the document. Later in 3. Results and discussion section you use a nomenclature for the rest 21 variables. You may consider adding nomenclature at some point of the document for these 21 variables.

2.6 Model construction: I suggest you should describe both structures a bit more. Figure 3 helps but a brief explanation will help.

Figure 3: I know some acronyms are conventional, but some should be established in the document.

Figure 4, 5 and 8: Variables are not readable, even zooming. Images should be presented differently. In Figure 5 you can not read values of the matrix.

Line 213: Which sizes were used?

Line 223: “complexity”. you may want to show some examples of the complexity to support this observation.

Author Response

Thank you for editor’ and reviewer’s opinions, these comments are very helpful to improve the quality of the manuscript. Now I response the reviewer’s comments with a point by point (in red) and highlight the changes in revised manuscript. Full details of the files are listed. We sincerely hope that you find our responses and modifications satisfactory and that the manuscript is now acceptable for publication.

Comment 1:

Abstract

Line 14: Typo “Abstract: Abstract:”

Line 19: Typo “construct-ed”.

Response 1:

We are very sorry for our careless mistakes and it were rectified at line 14 “Abstract:” and at line 23 “constructed”.

Comment 2:

Introduction

Line 31: Typo “ad-vantage”

Response 2:

We are very sorry for our careless mistake and it was rectified at line 33 “advantage”.

Comment 3: 

Figure 1: Left maps are not necessary, aerial images with experiment setup and coordinates are enough. It would be better to enhance the quality of figures and labels of the experiment setup.

Response 3:

We appreciate it very much for this good suggestion, and we have done it according to your ideas (Figure 1).

Figure 1. Overview of experimental site.

Comment 4: 

Line 111: Could you explain more how to get those 180 samples. I am confused with table 1. Last column does not sum 180. It is 60+30+90+30.

Response 4:

We are very sorry for our careless mistake and it was rectified at Table1. Data samples include 2 cotton varieties, 3 repeated experiments, 5 collection periods, 6 nitrogen gradients, a total of 180 (2×3×5×6) samples

Table 1. Data set division.

Variety

Dataset

June 28

July 6

July 16

July 28

August 7

In total

Xinluzao45

Training set

12

12

12

12

12

60

Test set

6

6

6

6

6

30

Xinluzao53

Training set

12

12

12

12

12

60

Test set

6

6

6

6

6

30

Comment 5: 

2.4 Feature extraction: Table 2 explains the 25 variables, and it sets on the variables columns the nomenclature used in the document. Later in 3. Results and discussion section you use a nomenclature for the rest 21 variables. You may consider adding nomenclature at some point of the document for these 21 variables.

Response 5:

We appreciate it very much for this good suggestion, and we have have added nomenclature to the rest 21 variables at lines 134–139.

Lines 134–139: “Texture features included mean, standard deviation (std), and angular second moment (Asm), entropy (Ent), contrast (Con), and inverse differential moment (Idm) under gray level co-occurrence matrix of 0°, 45°, 90°, and 135°. Hue (H), saturation (S), and intensity (I) were used in HSI color model to represent images, which was more in line with the way people describe and interpret colors.”

The rest 21 variables you mentioned in manuscript are hue (H), saturation (S), intensity (I), mean, standard deviation (std) and angular second moment (Asm), entropy (Ent), contrast (Con), and inverse differential moment (Idm) under gray level co-occurrence matrix of 0° (Asm0, Ent0, Con0, Idm0), 45° (Asm45, Ent45, Con45, Idm45), 90° (Asm90, Ent90, Con90, Idm90), and 135° (Asm135, Ent135, Con135, Idm135).

Comment 6: 

2.6 Model construction: I suggest you should describe both structures a bit more. Figure 3 helps but a brief explanation will help.

Response 6:

We appreciate it very much for this good suggestion, and we have have added explanation to CNN structures at lines 176–184.

Figure 3. CNN structures, (a) 1dcnn structure, (b) 2dcnn structure.

Lines 176-184: “One-dimension (1D) CNN (Figure 3(a)) included input layer, three 1D convolution layers (1D-CNN), full connection layer (FC) and output layer. 1D-CNN included 1D convolution structure (Conv1D), rectified linear unit activation function (ReLU), batch normalization (BatchNorm) and max pooling (MaxPool). The number of convolution kernels of the three 1D-CNN was 32, 64 and 128. Two-dimension (2D) CNN (Figure 3(b)) included input layer, four 2D convolution layers (2D-CNN), FC and output layer. 2D-CNN included 2D convolution structure (Conv2D), ReLU, BatchNorm and Max-Pool. The number of convolution kernels of the four 2D-CNN was 32, 64, 128 and 256.”

Comment 7: 

Figure 3: I know some acronyms are conventional, but some should be established in the document.

Response 7:

We appreciate it very much for this good suggestion, and we have have added explanation to acronyms (in Figure 3) at lines 176–184. These acronyms include one-dimension (1D) convolution layers (1D-CNN), two-dimension (2D) convolution layers (2D-CNN), full connection layer (FC), 1D convolution structure (Conv1D), 2D convolution structure (Conv2D), rectified linear unit activation function (ReLU), batch normalization (BatchNorm) and max pooling (MaxPool).

Lines 176-184: “One-dimension (1D) CNN (Figure 3(a)) included input layer, three 1D convolution layers (1D-CNN), full connection layer (FC) and output layer. 1D-CNN included 1D convolution structure (Conv1D), rectified linear unit activation function (ReLU), batch normalization (BatchNorm) and max pooling (MaxPool). The number of convolution kernels of the three 1D-CNN was 32, 64 and 128. Two-dimension (2D) CNN (Figure 3(b)) included input layer, four 2D convolution layers (2D-CNN), FC and output layer. 2D-CNN included 2D convolution structure (Conv2D), ReLU, BatchNorm and Max-Pool. The number of convolution kernels of the four 2D-CNN was 32, 64, 128 and 256.”

Comment 8: 

Figure 4, 5 and 8: Variables are not readable, even zooming. Images should be presented differently. In Figure 5 you can not read values of the matrix.

Response 8:

We appreciate it very much for this good suggestion, and we have replaced Figure 4, Figure 5 and Figure 8.

Figure 4. Importance of partial least squares variables, (a) Xinluzao 45, and (b) Xinluzao53.

Figure 5. Pearson correlation analysis, (a) Xinluzao 45, (b) Xinluzao 53.

Figure 8. 1D CNN saliency map, (a) Xinluzao 45, (b) Xinluzao 53. The horizontal axis was the 46 manual feature variables extracted in this study, and the vertical axis was the feature saliency obtained by saliency map method.

Comment 9: 

Line 213: Which sizes were used?

Response 9:

We appreciate it very much for this good question, the size you mentioned was a parameter that we needed to adjust when modeling, the range of this parameter is (2, 3, …, 23). We have added the range of size at lines 221-224.

Lines 221-224: “When modeling with the features screened by Pearson correlation analysis, the fea-tures were sorted according to the correlation strength, and LR and SVR models were constructed with the data sets of different sizes (2, 3, …, 23) to screen the optimal model.”

Comment 10: 

Line 223: “complexity”. you may want to show some examples of the complexity to support this observation.

Response 10:

We appreciate it very much for this good suggestion, and we have shown an example at lines 232-237.

Lines 232-237: “This was because when collecting UAV remote sensing data, the growth of Xinluzao 45 was better than that of Xinluzao 53, and the complexity of each plot in the image data of the latter was higher than that of the former. For example, compared with Xinluzao 45, Xinluzao 53 had less dense vegetation and more exposed bare soil (Figure 1), which brought difficulties to evaluate the leaf nitrogen level of the whole site.”

Figure 1. Overview of experimental site.

Revision date

21 July 2022

Reviewer 2 Report

The topic of the paper is fine which is the use of UAV RGB Images for predicting cotton leaf nitrogen content

I have many comments to the authors

Abstract: The abstract should be self explanatory (remove all abbreviation)

Line 15: define UAV

Line 16 gradient is replicated

Line 19: correct the temrs constructed

the end of the abstract should contain a cobnclusion and the limit of the present study

Introduction

Line 31: correct the terms advantage

Line 45: correct the terms extraction

Materials and methods

Lines 80-81: I suggest to correct the sentence as follows: avialable nitrogen, phosphorous and potassium content of ....., ..... and .......mg kg-1 respectively

Line 88: delete the terms with water

Line 104: replace to by until

Line 113: leave space before table 1

lines 150-151: improve the clarity and readability of the sentence

line 158: plz add coefficient

figure 6, 7, 8 and 9 should be self explanatory and footnotes should be inserted as well and values in the axis should be defined values of what

conclusions

line 259 correct the terms extraction

line 264 correct the terms features

line 266 correct the term engineering

the conclusion lacks the limit of the present study and the need for future research 

authors should there respond to the question where future works should focus

references 

plz check the refernces number 4, 5 and 10 and authors here should be consistent with the other references and the style of the journal

Author Response

Response to Reviewer 2 Comments

Thank you for editor’ and reviewer’s opinions, these comments are very helpful to improve the quality of the manuscript. Now I response the reviewer’s comments with a point by point (in red) and highlight the changes in revised manuscript. Full details of the files are listed. We sincerely hope that you find our responses and modifications satisfactory and that the manuscript is now acceptable for publication.

Comment 1:

The topic of the paper is fine which is the use of UAV RGB Images for predicting cotton leaf nitrogen content.

Response 1:

We thank you very much for these positive and encouraging comments. In particular, we really appreciate your insightful suggestions.

Comment 2:

Abstract: The abstract should be self explanatory (remove all abbreviation).

Response 2:

We appreciate it very much for this good suggestion, and we have removed all abbreviation at lines 14-27.

Lines 14-27: “Abstract: Rapid and accurate prediction of crop nitrogen content is of great significance for guiding precise fertilization. In this study, unmanned aerial vehicle (UAV) digital camera was used to collect cotton canopy RGB images at 20 m height, and 2 cotton varieties and 6 Nitrogen gradients were used to predict nitrogen content in the cotton canopy. After image-preprocessing, 46 hand features are extracted and deep features are extracted by convolutional neural network (CNN). Partial least squares and Pearson are used to feature dimensionality reduction respectively. The linear regression, support vector machine, and one-dimension CNN regression models were constructed with manual features as input, and the deep features were used as input to construct two-dimension CNN regression model to achieve accurate prediction of cotton canopy nitrogen. It is verified that the manual feature and deep feature models constructed from UAV RGB images have good pre-diction effects. R2 = 0.80 and RMSE = 1.67 g kg-1 of Xinluzao 45 optimal model and R2 = 0.42 and RMSE = 3.13 g kg-1 of Xinluzao 53 optimal model. The results show that the UAV RGB image and machine learning technology can be used to predict the nitrogen content of large-scale cotton, but due to the insufficient data samples, the accuracy and stability of the prediction model still need to be improved.”

Comment 3:

Line 15: define UAV.

Response 3:

We appreciate it very much for this good suggestion, and we have defined at line 15.

Unmanned aerial vehicle (UAV).

Comment 4:

Line 16 gradient is replicated.

Response 4:

We are very sorry for our careless mistake and it was deleted at line 16.

Comment 5:

Line 19: correct the temrs constructed.

Response 5:

We are very sorry for our careless mistake and it was rectified at line 20 “constructed”.

Comment 6:

the end of the abstract should contain a conclusion and the limit of the present study.

Response 6:

We appreciate it very much for this good suggestion, and we have added conclusion and limit of the present study at lines 25-27.

Lines 25-27: “The results show that the UAV RGB image and machine learning technology can be used to predict the nitrogen content of large-scale cotton, but due to the insufficient data samples, the accuracy and stability of the prediction model still need to be improved.”

Comment 7:

Introduction

Line 31: correct the terms advantage.

Line 45: correct the terms extraction.

Response 7:

We are very sorry for our careless mistakes and it were rectified at line 33 “advantage” and at line 46 “extraction”.

Comment 8:

Materials and methods

Lines 80-81: I suggest to correct the sentence as follows: avialable nitrogen, phosphorous and potassium content of ....., ..... and .......mg kg-1 respectively.

Response 8:

We appreciate it very much for this good suggestion, and we have revised the sentence at lines 80-82.

Lines 80-82: “The soil type of the test site is irrigated gray desert soil, with organic matter, available nitrogen, available phosphorus and available potassium content of 11.75 g kg-1, 0.45 mg kg-1, 0.39 mg kg-1 and 375.21 mg kg-1.”

Comment 9:

Line 88: delete the terms with water.

Response 9:

We appreciate it very much for this good suggestion, and we have deleted the terms with water at line 88.

Comment 10:

Line 104: replace to by until.

Response 10:

We appreciate it very much for this good suggestion, and we have replaced to by until at line 104.

Comment 11:

Line 113: leave space before table 1.

Response 11:

We appreciate it very much for this good suggestion, and we have left space before table 1 at line 115.

Comment 12:

lines 150-151: improve the clarity and readability of the sentence.

Response 12:

We appreciate it very much for this good suggestion, and we have improved the clarity and readability of the sentence at lines 152-154.

Lines 152-154: “Feature selection realized data dimensionality reduction by selecting some of the most representative and good-performance features from the original features.”

Comment 13:

line 158: plz add coefficient.

Response 13:

We appreciate it very much for this good suggestion, and we have added coefficient at line 160.

Comment 14:

figure 6, 7, 8 and 9 should be self explanatory and footnotes should be inserted as well and values in the axis should be defined values of what.

Response 14:

We appreciate it very much for this good suggestion, and we have added footnotes to figure 6, 7, 8 and 9.

Figure 6. Prediction results, (a)-(c): Prediction results of Xinluzao 45 (LR, SVR, and PLSR); (d)-(f): Prediction results of Xinluzao 53 (LR, SVR, and PLSR). The horizontal axis and vertical axis were the factual and model predicted nitrogen content values of cotton respectively.

Figure 7. 1D CNN prediction results, (a) Xinluzao 45, (b) Xinluzao 53. The horizontal axis and vertical axis were the factual and model predicted nitrogen content values of cotton respectively.

Figure 8. 1D CNN saliency map, (a) Xinluzao 45, (b) Xinluzao 53. The horizontal axis was the 46 manual feature variables extracted in this study, and the vertical axis was the feature saliency obtained by saliency map method.

Figure 9. 2D CNN prediction results, (a) Xinluzao 45, (b) Xinluzao 53. The horizontal axis and vertical axis were the factual and model predicted nitrogen content values of cotton respectively.

Comment 15:

conclusions

line 259 correct the terms extraction.

line 264 correct the terms features.

line 266 correct the term engineering.

Response 15:

We are very sorry for our careless mistakes and it were rectified at line 275 “extraction”, at line 280 “features” and at line 282 “engineering”.

Comment 16:

the conclusion lacks the limit of the present study and the need for future research.

authors should there respond to the question where future works should focus.

Response 16:

Thanks for your comment on the conclusion, we have supplemented the content about the limit of the present study and the need for future research at lines 283-288.

Lines 283-288: “Therefore, it is feasible to predict cotton nitrogen content using UAV-based RGB remote sensing images, but it is necessary to further improve the generalization of prediction models for different varieties. In the next research, we should pay attention to more cotton varieties, planting areas and growth periods, continue to expanding data sam-ples and improving the cotton nitrogen content prediction model, and try to apply this method to the nutrition detection of other crops..”

Now, I will respond to the question where future works should focus. In the present study, the accuracy and stability of cotton nitrogen content prediction model still need to be improved. We will solve this problem from two aspects: (1) continue to expanding the data set, (2) continue to mining deep features and try to fuse deep features and manual features to improve modeling performance.

Comment 17:

references

plz check the refernces number 4, 5 and 10 and authors here should be consistent with the other references and the style of the journal.

Response 17:

We are very sorry for our careless mistakes and it were checked at line 306, 308 and 320.

Revision date

21 July 2022
